# Role of Herbal Teas in Regulating Cellular Homeostasis and Autophagy and Their Implications in Regulating Overall Health

**DOI:** 10.3390/nu13072162

**Published:** 2021-06-23

**Authors:** James Michael Brimson, Mani Iyer Prasanth, Dicson Sheeja Malar, Rajasekharan Sharika, Bhagavathi Sundaram Sivamaruthi, Periyanaina Kesika, Chaiyavat Chaiyasut, Tewin Tencomnao, Anchalee Prasansuklab

**Affiliations:** 1Natural Products for Neuroprotection and Anti-Ageing Research Unit, Chulalongkorn University, Bangkok 10330, Thailand; James.b@chula.ac.th (J.M.B.); prasanth.m.iyer@gmail.com (M.I.P.); sheeja.malar@gmail.com (D.S.M.); 2Department of Clinical Chemistry, Faculty of Allied Health Sciences, Chulalongkorn University, Bangkok 10330, Thailand; 3Department of Transfusion Medicine and Clinical Microbiology, Faculty of Allied Health Sciences, Chulalongkorn University, Bangkok 10330, Thailand; sharikarpillai@gmail.com; 4Innovation Center for Holistic Health, Nutraceuticals, and Cosmeceuticals, Faculty of Pharmacy, Chiang Mai University, Chiang Mai 50200, Thailand; Sivamaruthi.b@cmu.ac.th (B.S.S.); p.kesika@gmail.com (P.K.); chaiyavat@gmail.com (C.C.); 5College of Public Health Sciences, Chulalongkorn University, Bangkok 10330, Thailand

**Keywords:** herbal tea, phytotherapy, phytochemicals, autophagy, oxidative stress, inflammation, cancer, neurodegenerative diseases, metabolic disorders

## Abstract

Tea is one of the most popular and widely consumed beverages worldwide, and possesses numerous potential health benefits. Herbal teas are well-known to contain an abundance of polyphenol antioxidants and other ingredients, thereby implicating protection and treatment against various ailments, and maintaining overall health in humans, although their mechanisms of action have not yet been fully identified. Autophagy is a conserved mechanism present in organisms that maintains basal cellular homeostasis and is essential in mediating the pathogenesis of several diseases, including cancer, type II diabetes, obesity, and Alzheimer’s disease. The increasing prevalence of these diseases, which could be attributed to the imbalance in the level of autophagy, presents a considerable challenge in the healthcare industry. Natural medicine stands as an effective, safe, and economical alternative in balancing autophagy and maintaining homeostasis. Tea is a part of the diet for many people, and it could mediate autophagy as well. Here, we aim to provide an updated overview of popular herbal teas’ health-promoting and disease healing properties and in-depth information on their relation to autophagy and its related signaling molecules. The present review sheds more light on the significance of herbal teas in regulating autophagy, thereby improving overall health.

## 1. Introduction

### 1.1. What Is Autophagy

Macro-autophagy, henceforth referred to as autophagy, is a genetically conserved mechanism that degrades cellular organelles and proteins in times of stress or removes unwanted, damaged, or surplus items. Autophagy thus both provides a mechanism to protect against cellular starvation and acts to maintain cellular homeostasis. Autophagy may also be subdivided into different categories of selective autophagy, such as autophagy targeting the mitochondria (mitophagy) and chaperone-mediated autophagy [1].

Autophagy is characterized by the formation of a double membraned vesicle known as the autophagosome. The autophagosome will merge with a lysosome to form the autolysosome, where the contents will be degraded and recycled [2]. Autophagy or autophagic flux may be monitored by following the metabolism of LC3-I cleavage to LC3-II and the accumulation of LC3-II in the autophagosome. An apparent elevation of LC3-II relative to LC3-I visualized via Western blot or as puncta staining in the autophagosomes via confocal microscopy tends to suggest the activation of autophagy [3]. However, some autophagy inhibitors inhibit the binding of the autophagosome with the lysosome, which would lead to an apparent increase in LC3-II expression/staining. It is therefore essential to carry out all experiments in the presence and absence of autophagy inhibitors (3-MA or chloroquine) and activators (rapamycin/starvation) in order to provide context to the observed results [4]. 

Autophagy is controlled by autophagy-related genes (*Atg*). Two primary gatekeepers of autophagy are the protein kinases mechanistic target of rapamycin (mTOR) and 5’ AMP-activated protein kinase (AMPK). These two kinases work to negatively regulate autophagic activity through the phosphorylation of Unc-51-like kinases (ULK1 and ULK2 (*Atg*-1)) [4,5,6]. Autophagy induction by inhibiting mTOR using a drug such as rapamycin results in the dephosphorylation and activation of the ULK kinases. Subsequently, ULK phosphorylates, and thus activates, Beclin-1 (*Atg*-6). Beclin-1 forms part of a complex known as the autophagy-inducible Beclin-1 complex, which also contains other proteins, including PIK3R4 (*Atg*-14) and PIK3C3. Once activated, the ULK-Beclin-1 complex migrates to the autophagosome, activating downstream autophagic functions [7]. There are many other proteins and receptor-regulated pathways and stress conditions that may lead to the activation of various autophagic pathways, such as sigma-1 receptor activation [8], inflammation [9], Huntingtin [10], Toll-like receptors [11], and G-protein-coupled receptors [12].

### 1.2. Why Is Autophagy Important

As well as its role in various disease states (discussed below), autophagy has several roles and cellular functions essential for maintaining basal homeostasis. In nutrient starvation, autophagy is activated to degrade surplus or unrequired proteins into their constituent amino acids, which can be recycled to produce proteins that are essential for survival [7]. Much of the early work describing autophagy was carried out on budding yeast strains, which activated a strong autophagic response under starvation conditions. Furthermore, it was shown that the inhibition of various *Atg* genes resulted in a reduction in the viability of yeast cells subjected to starvation conditions. One such *Atg* gene is *Atg*-7, which appears to be important in developing the autophagosomal membrane and is essential to cellular degradation and recycling. *Atg*-7-deficient mice have impaired starvation-induced autophagy [7]. Concerning nutrient starvation, autophagy is also crucial to muscle homeostasis during physical exercise. Exercise-induced autophagy has been well documented, with the first studies noting increased size and number of autophagic vacuoles in exercised animals’ liver and skeletal muscle [13]. More recent studies have shown the importance of autophagy in acute exercise, with autophagy being upregulated during acute exercise, while a mouse model lacking collagen VI causing autophagy inhibition [14] is intolerant to exercise [15]. Interestingly, there have also been reports of the upregulation of autophagy in the brain after exercise, which may well explain, at least in part, some of the benefits of long-term exercise of neurological health and the prevention of neurodegenerative disease [16,17,18]. 

### 1.3. The Role of Autophagy in Disease

#### 1.3.1. Cancer

Autophagy predominately acts in a cytoprotective manner during times of stress. Therefore, it is not surprising that autophagic pathways become co-opted by cancer cells to avoid apoptosis. The role of autophagy in tumorigenesis is somewhat paradoxical; on the one hand, loss of the gene essential to autophagy, Beclin-1(*Atg*-6), is found in many tumors, including breast, ovarian, and prostate cancers. Furthermore, autophagy-defective Beclin-1 mice and *Atg*-4C deficient mice are predisposed to tumor growth. On the other hand, there is much evidence supporting the role of autophagy activation in tumor generation, protecting cells from hypoxia inside the tumor before any angiogenesis takes place in the tumor development.

The basal level of autophagy acts as a tumor suppressor via the reduction in cellular stress by removing damaged proteins and cellular components, allowing for the regulation of cellular homeostasis. For example, damaged mitochondria induce a large amount of oxidative stress if not swiftly taken care of, resulting in further cellular damage and the promotion of carcinogenesis [19,20,21]. The autophagy gene Beclin-1 acts as a tumor suppressor, and interference in its expression can lead to tumor development [22]. Furthermore, it has been shown that Beclin-1 deficiency is associated with increased angiogenesis when melanoma tumor cell lines are subjected to hypoxic conditions [23] and that Beclin-1 expression disrupts the growth of colon cancer cells [24]. 

Many cancer cells rely on autophagy activation for survival, and in many ways, they are more reliant on autophagy than normal cells. This may be due to changes in the microenvironment (such as hypoxic regions of a tumor) that induce metabolic stress and demands on the cell caused by unregulated proliferative activity. In RAS-transformed tumor cells, autophagy is upregulated and induces tumor growth and survival [25,26,27]. 

One way of thinking about autophagy is the “Goldilocks scenario”, where too little autophagy results in susceptibility to cellular death in response to stress, whereas too much autophagy leads to cell death through excessive self-digestion. In the middle, there is an amount of autophagic activity that is just right for cellular homeostasis. Therefore, one might argue that activating or preventing autophagy may be a suitable treatment for cancer cells. However, both approaches can be risky if the cells do not subsequently die. Treatments to reduce autophagy might remove some tumor-suppressive function of autophagy, whereas increasing autophagy might induce further tumorigeneses and increased survival of tumor cells [28].

#### 1.3.2. Neurodegenerative Disease

Many studies have indicated that autophagic activity and the components required for autophagy diminish with aging [29,30], and it is well known that aging is one of the critical risk factors for the development of neurodegenerative disease. Autophagy plays a crucial role in clearing the build-up of amyloid-β and hyperphosphorylated tau protein and removing damaged organelles such as the mitochondria, which are left to degrade, releasing ROS signals for cell death. Many studies have shown that upregulating autophagy results in a decrease in amyloid-β expression [31,32,33,34]. However, as with cancer (as discussed above), autophagy has a paradoxical relationship with Alzheimer’s disease, as other studies show that increased autophagy results in an increase in amyloid-β in the extracellular space [35], with the autophagosomes generating the amyloid-β from the amyloid precursor protein [31,36]. 

Parkinson’s disease (PD) is characterized by the loss of dopaminergic neurons in the substantia nigra region of the brain and the formation of α-synuclein inclusions. There is compelling evidence that PD has a link to autophagy dysfunction, as the α-synuclein inclusions have been shown to change autophagic function [37], resulting in decreased lysosome and autophagosome fusion and a reduction in protein degradation. Furthermore, early-onset PD studies have shown genetic mutations in phosphatase and tensin homolog-induced putative kinase 1 (PINK1) and E3 ubiquitin ligase (Parkin). Both parkin and PINK1 have been shown to activate autophagy and target mitochondria for destruction as part of the mitophagy pathway [38,39,40,41]. 

#### 1.3.3. Metabolic Disorders

Obesity and type II diabetes are worldwide health problems. Autophagy has been linked to the pathology of both. One of the critical aspects appears to be the destruction and removal of malfunctioning mitochondria as part of the mitophagy process. Insulin resistance and β-cell destruction are dependent on the damage to mitochondria caused by metabolic overload, leading to incomplete β-oxidation, oxidative stress, further mitochondrial damage, and the build-up of toxic lipid intermediates. The removal of these damaged mitochondria via mitophagy eliminates this cycle of oxidative stress and reverses the pathogenic process [42]. 

Studies have shown that β-cells in diabetic organ donors show altered autophagy. Furthermore, the study showed that while the genes involved in the early stages of autophagy, *Beclin*-1 and *Atg*1, appeared unaltered in expression, genes involved in the later stages of autophagy *LAMP*-2 and *cathepsin B* and *D* showed a significant reduction in expression. This, along with the large vacuole overload associated with the donor β-cells, suggested that incomplete autophagy potentially played a role in the β-cell toxicity [43].

#### 1.3.4. Inflammatory Stress

Aside from autophagy, many other cellular mechanisms deal with stress. One such mechanism is inflammation, which can be both cytoprotective and detrimental to cell survival depending on the circumstance and stresses involved [44]. There are several links between autophagy and inflammation in the pathology of many diseases involving inflammation [45]. Notably, under physiological and pathological conditions, reciprocal crosstalk and context-dependent regulation have been observed between IKK/NF-κB signaling and autophagy [46,47]. Although most of the studies show an inverse relation [48,49,50,51,52], there are also reports on the activation of autophagy by NFκB [53,54,55], which may depend on the different inflammatory response, whether acute or chronic, types of cells and pathological conditions [47]. For instance, in neurodegenerative diseases, suppression of autophagy via NF-κB activation causes cell death [56], while the same in cancer results in cell survival [57]. In addition to the IKK/NF-κB axis, IKK alone can also trigger autophagy in an NF-κB-independent manner [58,59]. IKK/NF-κB activates autophagy by inducing the expression of autophagy-related genes and proteins (Beclin-1, *Atg*-1, *Atg*-5, LC3) [58,59,60,61] or inhibit autophagy by inducing autophagy repressors (A20, mTOR) [62,63,64] and suppressing autophagy inducers (JNK, p53, ROS) [51,65,66]. In addition, an interplay between the pro-inflammatory cytokines (IFN-γ, TNF-α. IL-1β) and autophagy [67,68,69] through various mechanisms, including JNK, JAK1/2, ERK1/2, PI3K, p38, and MAPK was also observed under various stimulatory conditions [70,71,72,73,74].

## 2. Herbal Teas and Autophagy

Tea, the most consumed beverage worldwide, has an abundance of antioxidants in it [75]. The tea leaves are harvested, dried, and steamed, which prevents the activation of polyphenol oxidizing enzymes, thereby protecting tea’s nutritional value [75,76]. Tea possesses significant antioxidative, anti-inflammatory, antimicrobial, anticarcinogenic, antihypertensive, neuroprotective, cholesterol-lowering, and thermogenic properties [77]. Tea consumption can modulate microbial diversity and the ratio of Firmicutes to Bacteroidetes in the gut microbiome, thereby aiding to offset dysbiosis triggered by obesity or high-fat diets [78]. Consumption of tea was reported to reduce the possibility of developing tuberculosis [79].

Tea can be prepared from different plants, which are collectively called “herbal tea”. In this review, we discuss different kinds of tea produced from *Camellia sinensis* (green, oolong, and black tea), *Carthamus tinctorius* L. (safflower tea), *Zingiber officinale* (tea with ginger extract), *Hibiscus sabdariffa* L. (hibiscus tea), *Cymbopogon citratus* (lemongrass tea), *Rhinacanthus nasutus* L. (Rhinacanthus tea) and *Jasminum* sp. (jasmine tea), their health benefits and their relation to autophagy.

### 2.1. Camellia Sinensis

One of the most prevalent tea plants produced and utilized all around the world is *Camellia sinensis*. Based on the level of antioxidants present and the degree of fermentation, tea prepared from *C. sinensis* has been classified into green tea, oolong tea, and black tea. These teas are categorized according to the oxidization levels (low for green tea, intermediate for oolong tea, and high for black tea) with varied contents, and they contain low levels of chlorogenic acids as well as other substances such as flavonoids (high for green tea, intermediate for oolong tea, and low for black tea) [75,80]. Processing tea leaves under anoxic conditions using nitrogen leads to a 10 to 20 times increase in GABA (the major inhibitory neurotransmitter in the brain), leading to GABA-enriched green, oolong, and black tea [81]. 

#### 2.1.1. Green Tea

Green tea possesses the maximum level of antioxidants, which aids in imparting antiaging [82] and neuroprotective effects [83,84]. It can also suppress several diseases such as cancer [85], cardiovascular conditions [86,87], and obesity [88], inhibit tooth decay and reduce blood pressure, along with exerting antibacterial, antioxidant, and antidiabetic properties [83,89,90]. Green tea mainly consists of polyphenols (~90%), amino acids (~7%), theanine, proanthocyanidins, and caffeine (~3%). Catechin (C), epicatechin (EC), gallocatechin (GC), epigallocatechin (EGC), epicatechin gallate (ECG), epigallocatechin gallate (EGCG), and gallocatechin gallate (GCG) are the significant polyphenol catechins present [76,91]. Among these, EGCG is the most abundant polyphenol, comprising almost 50–80% of total polyphenols, and is responsible for the majority of health benefits [91].

Autophagy could protect the host’s overall health, thereby reducing the severity of conditions such as cardiomyopathy, cancer, and neurodegeneration [92,93]. It can play a critical role in the occurrence and development of many human diseases, including cancer, neurological diseases, diabetes, cardiovascular diseases, and injury [94]. The different beneficial pathways of the host, such as the mitogen-activated protein kinase (MAPK) pathway, inhibition of the mammalian target of rapamycin (mTOR), PI3-kinase-mediated FOXO transcription, Nrf-2 transcription factor, and calorie restriction, can activate autophagy [95,96,97]. 

Autophagy plays a critical role in modulating the overall health benefits of green tea. For instance, tea polyphenols were reported to activate autophagy through the mTOR pathway, thereby delaying apoptosis upon endoplasmic reticulum stress in HEK293T cells [98]. Green tea activated autophagy in HL-60 xenografts by increasing the activity of PI3 kinase and Beclin-1 [96] in primary neuronal cells by inducing sirtuins [99]. In *Caenorhabditis elegans,* green tea could synergistically extend the overall lifespan mediated by *daf*-16 and *skn*-1 [82], activating autophagy. In hepatic cells, EGCG was observed to activate AMPK, thereby activating autophagy [100]. Tea polyphenols were able to activate autophagy in high fat-fed rats and inhibit the level of high blood glucose-induced autophagy [101]. High glucose was reported to reduce autophagy and increase apoptosis in Muller cells, which was reversed by EGCG [102]. EGCG alone or combined with exercise-mediated SIRT1 and PGC-1α expressions, Nrf-1, Nrf-2, HO-1, and TFAM protein expressions, increased mitochondrial fusion protein expression in the hippocampus and increased the expression of autophagy-related genes including NLX and BNIP3, thereby ameliorating ischemic injury [103]. An increase in oxidative stress and reduced autophagy are the hallmarks of osteoarthritis and sarcopenia [104]. EGCG was reported to reduce chondrocyte apoptosis during osteoarthritis in rat models by activating autophagy [105]. Green tea supplementation could effectively suppress the activation of autophagy inhibitors, thereby reducing muscle damage [106] and reducing ROS accumulation [107]. 

Autophagy plays a significant role in the anti-cancer properties of green tea [108]. The activation of autophagy by green tea and its polyphenols inhibited the growth of breast cancer cells [109] and brain tumor cells [110]. The transcription factor Nrf-2 can either suppress or promotes tumorigenic properties of cells depending on the intracellular location, hyperactivation, cancer types, and stages [111]. Though the antioxidant activity mediated by Nrf-2 plays a critical role in mediating the survival of breast cancer cells via autophagy, a reversal of effect was observed in colorectal cancer cells [93,112]. EGCG was found to increase the specificity and sensitivity of radiation in targeting cancer cells through the Nrf-2 mediated autophagy mechanism in colorectal cancer cells [93]. The chemotherapeutic drug doxorubicin, along with EGCG, synergistically improved the clinical effect in treating osteosarcoma cancer cells [97]. The same combination was observed to have chemo-photothermal synergistic therapy in a mouse HeLa tumor model by inducing autophagy [113]. The prevention and treatment of hepatocellular carcinoma in HepG2 cells were initiated by EGCG by regulating α-fetal protein (AFP) secretion, thereby modulating autophagy [114]. AFP is a major plasma protein and serves as a biomarker for hepatocellular carcinoma [115]. EGCG induced LC3-II production by inhibiting dimerization of LC3-I, and promoted autophagic degradation of AFP [114]. The anti-cancer activity in colon cancer cells by EGCG [116] and tea polysaccharides [117] is also mediated by the activation of autophagy.

Impaired autophagy has been linked with several neurodegenerative diseases [118]. EGCG was reported to activate Nrf-2 in SH-SY5Y neuroblastoma cells activating autophagy and increased adaptor proteins NDP52, and p62 mRNA levels, thereby mediating neuroprotection [119]. EGCG could prevent the human vascular endothelial cells from oxidative stress by activating autophagy via the mTOR pathway, thereby protecting them from cardiovascular diseases [120]. Rats fed with EGCG could reduce the toxicity and oxidative stress induced by acetaminophen in the liver by activating autophagy, thereby providing hepatoprotection [121]. EGCG also has cytoprotective roles in cancer treatment, protecting salivary glands during radiation therapy for mouth cancers, although autophagy activation and inhibition were not investigated [122]. The anti-obesity effect of EGCG in high fat-fed mice was dependent on Beclin-1 mediated autophagy [123]. EGCG could activate autophagy inside mice macrophage cells, thereby eliminating tuberculosis microbes [124].

However, higher concentrations of EGCG can inhibit autophagy, which could also lead to apoptosis [125]. The autophagy activation properties of green tea majorly depend upon the dosage of polyphenol, the level of stress, and the cells involved [126]. On the other hand, some studies suggest that EGCG can reduce the activation of autophagy, which is beneficial to the host. For example, green tea polyphenols resisted autophagy levels in PC12 cells activated by hydrogen peroxide [127]. In another study, tea polyphenols were reported to rescue the induction of autophagy by a plasticizer, tri-ortho-cresyl phosphate, which can cause female reproductive damage [128].

#### 2.1.2. Oolong Tea

Oolong tea is considered the most efficient tea in mediating weight loss by altering the bile acid metabolism or lipid metabolism mediated by short-chain fatty acids [129,130]. The polysaccharides and polyphenols of oolong tea were found to suppress fat accumulation, reduce serum leptin levels and improve antioxidant levels in high-fat-fed rats. This is carried out through fatty acid biosynthesis pathways, steroid hormone biosynthesis, unsaturated fatty acid biosynthesis, fatty acid elongation, glycerolipid metabolism, and glycerophospholipid metabolism [131].

Consumption of oolong and black tea were observed to be beneficial in a study conducted in subjects aged over 60, as the overall health and lifestyle were better than non-tea drinkers, and the problems related to mobility, pain, discomfort, anxiety, and depression were lesser [132]. Tea (oolong or green or black) consumed in three or more cups per day could prevent the worsening of Geriatric Depression Scale symptoms of depression and reduce the likelihood of depression [133]. Green and oolong tea might also reduce the chances of hyperhomocysteine in subjects with hypertension [134]. GABA-enriched and regular oolong tea significantly reduced acute stress and altered heart rate variability, leading to reduced stress response, thereby acting as a neuro-nutraceutical [135]. In another study, a cup of GABA-fortified oolong tea was reported to significantly decrease the immediate stress score and improve heart rate variability in 30 subjects using a pre-post cohort study design [81]. It also influenced behavioral parameters linked to depression in the mouse model of post-stroke depression [136]. Oolong tea exhibits cytoprotective effects in cardiomyocytes and H9c2 cells against hypoxia by suppressing the JNK, enhancing Nrf-2 mediated antioxidant mechanism, and IGFIR/p-Akt associated survival-mechanism [137]. Similar results were also observed with isoproterenol-induced toxicity in H9c2 cells. The extract could increase cell viability and reduce apoptosis by reducing caspase-3 and cytochrome c, blocked hypertrophy markers calcineurin, NFATc3, and BNP, increasing cell proliferation markers -PI_3_K and AKT [138].

Habitual tea consumption can reduce the risk of neurocognitive disorders [139]. The oolong tea extract in the form of dietary supplement “percepta” was able to reduce Aβ fibrils and tau protein tangles in vitro, as it caused a marked inhibition of beta-sheet secondary folding of tau protein into paired helical filaments [140]. Oolong tea extracts had a protective effect against glutamate-induced cell death in Neuro-2A and HT-22 cells. The extracts reduced intracellular reactive oxygen species accumulation and induced gene expression of cellular antioxidant enzymes such as GPx, GSTs, and SODs. These extracts also increased the average neurite length and induced the expression of GAP-43 and Ten-4 involved in neural development in Neuro-2a cells. Moreover, they had protective effects against Aβ-induced paralysis, chemotaxis deficiency, and α-synuclein aggregation in *C. elegans* [141]. Teaghrelin, a compound in oolong tea, activated AMP-activated protein kinase (AMPK)/sirtuin 1(SIRT1)/peroxisome proliferator-activated receptor gamma (PPARγ) coactivator-1α (PGC-1α) and extracellular signal-regulated kinases 1 and 2 (ERK1/2) pathways in SH-SY5Y cells to antagonize mitochondrial toxin, MPP^+^-induced cell death [142]. AMPK activation can maintain cellular homeostasis and offers neuroprotection by inhibiting mTORC1 and phosphorylating Ulk-1to initiate autophagy [143,144].

Oolong tea in synergistic combination with selenium could induce anti-proliferative effects by increasing intracellular ROS production, G2/M phase arrest, inducing expressions of the p53, Bax, caspase 3, and reducing Bcl-2 and CDK2, leading to cell apoptosis [145]. Oolong tea can significantly induce apoptosis of MDA-MB231 cells, mainly through the death-receptor-mediated extrinsic apoptotic pathway [146]. It can induce DNA damage and cleavage and play an inhibitory role in breast cancer cell growth, proliferation and tumorigenesis, thereby acting as a chemo-preventive agent against breast cancer [147]. The phenolics-rich fraction of oolong tea had more potent antioxidant activity, inhibiting the growth, migration, and invasion of breast cancer cells [148]. Fermentation of oolong tea using a fungus, *Aspergillus sojae*, can improve the gallic acid content of the tea extract, which could enhance the demethylation effects and a significant reduction in the nuclear abundances of DNMT1, DNMT3A, and DNMT3B in lung cancer cell lines [149]. Studies reported a negative regulation of autophagy by DNA methyl transferases that could favor the proliferation of cancer cells [150,151,152]. Hence, inhibition of DNMTs by oolong tea can induce autophagy and inhibit tumorigenesis.

The polyphenols and polysaccharides of oolong tea synergistically act on hepatocellular carcinoma cells. The combination was also able to enhance the antioxidant and immune levels in mice models [153]. An EGCG dimer, theasinensin A, was able to protect mice from liver injury mediated by carbon tetrachloride. It reduced collagen deposition and suppress hepatic α-smooth muscle actin and matrix metallopeptidase-9 expression through the inhibition of transforming growth factor β, thereby protecting against liver fibrosis [154]. Theasinensin A treatment administered to human U937 cells showed a cytotoxic effect by loss of mitochondrial membrane potential, an increase in ROS and proteolytic cleavage of poly(ADP-ribose) polymerase (PARP) [155].

Regular tea consumption (green, oolong, or black) is associated with a reduced risk of ovarian cancer [156]. Catechins of oolong tea alleviate polycystic ovary syndrome (PCOS) in mice induced by insulin combined with human chorionic gonadotropin. The catechins reduced E2, FSH, and LH levels in the blood and the ratio of LH/FSH, downregulated MMP-2, MMP-9, and p-NF-κB p65 in the blood the uterus and improve glucose metabolism and insulin resistance [157]. Signal networking analysis has shown the potential links of several genes, including NF-κB activation and autophagy-related genes during PCOS. Tissue analysis of individuals with PCOS and animal models also show induction of autophagy, which was observed in the increase in expression of LC3B-II and Beclin-1 [158]. Polyphenols in oolong tea can exhibit anti-osteoclastogenic activity by inhibiting receptor activator of nuclear factor-κB-mediated p38 activation, thereby preventing bone diseases [159]. Oolong tea can increase calcaneus bone mineral density in postmenopausal women, thereby reducing the chances of osteoporosis [160]. Oolong tea extract decreased mRNA levels of vascular inflammatory markers such as tissue necrosis factor-alpha (TNF-α), vascular cell adhesion molecule-1 (VCAM-1), and E-selectin, which was otherwise increased by carnitine consumption in mice. The oolong tea extract was thus able to protect the host from vascular inflammation [161]. Inhibition of inflammatory markers, including NF-κB, may act beneficially in modulating the detrimental effects of autophagy during PCOS, osteoporosis, and other related diseases. 

#### 2.1.3. Black Tea

Among the three different teas from *C. sinensis*, the most fermented tea version is black tea. The rate of fermentation will reduce the amount of polyphenol oxidase in the tea. In other words, green tea contains the maximum level of antioxidants, whereas oolong and black tea have comparatively lower levels [162,163]. However, black tea is the most consumed tea compared to green and oolong tea and there is insufficient evidence suggesting that green tea truly has better health-promoting properties than black tea [164]. The significant phenolics present in black tea are theaflavin and theasinensin, along with theaflagallins, theaflavates, theaflavic acids, theacitrins, and dehydrodicatechins.

Black tea will prevent the growth and propagation of *Bifidobacterium* spp and promote the *Akkermansia* spp in the gut, while green tea could facilitate the development of both [165]. Both green and black tea altered the proportions of a wide range of intestinal microbes in obese mice and exhibited similar gut microbiota profiles but significantly different gut microbiota profiles without tea intervention [166]. The modification of intestinal microbiota induced by tea phenolics may further contribute to the integrity of the mucus layer, attenuate the inflammatory process, regulate metabolic disorders, underlining the prebiotic potential of black tea phenolics [164].

Black tea polyphenols involve the activation of Nrf-2 through phosphorylation by PKC and PI3-kinases in hepatic cells, which is critical for the increased stability of Nrf-2 upon release from Keap1 and nuclear translocation, respectively [164,167]. Both black and green tea polyphenols could prevent the development of hypertension and blood pressure in stroke-prone spontaneously hypertensive rats, which could be attributed to its antioxidant potential. They could also prevent the phosphorylation of the myosin light chain in the aorta [168].

Theaflavins have been shown to inhibit the proliferation, survival, and migration of many cancer cells while promoting apoptosis. Treatment with theaflavins has been associated with increased levels of cleaved poly(ADP-ribose) polymerase (PARP) and cleaved caspases-3, -7, -8, and -9, all markers of apoptosis, and increased expression of the proapoptotic marker Bcl-2-associated X protein (Bax) and concomitant reduction in the antiapoptotic marker B-cell lymphoma 2 (Bcl-2). Additionally, theaflavin treatment reduced phosphorylated Akt, phosphorylated mechanistic target of rapamycin (mTOR), phosphatidylinositol 3-kinase (PI3K), and c-Myc levels with increased expression of the tumor suppressor p53 [169]. Theaflavins can induce apoptosis via p53 in breast cancer cells [170].

Theaflavins could increase the expression levels of proteins related to autophagy and connexin in neonatal cardiomyocytes with high glucose. It could also restore the normal functioning of gap-junctional intercellular communication and activated phosphorylated AMPK. It is believed that theaflavins partly reversed the inhibition of connexin expression and autophagy induced by high glucose in neonatal cardiomyocytes, partly by restoring AMPK activity. Inhibition of autophagy might be protective by preserving connexin expression in cells stimulated by high glucose [171].

### 2.2. Safflower Tea

*Carthamus tinctorius* L, commonly known as safflower, is a commercially important annual or winter annual herbaceous thistle-like plant and reported to be grown in 60 countries. It belongs to the Asteraceae family. As a drought-resistant plant, it is tolerant and adaptive to various soil and climatic conditions. The whole plant is used as a forage for ruminating animals due to its high nutritive value. The use of safflower dates back several hundreds of years, as seeds and floret garlands were also found in Egypt’s mummified bodies [172,173,174]. 

Despite being a multipurpose plant, whose seeds and flowers are also rich in health benefits and have been used in traditional medicine for a long time, safflower is chiefly cultivated for its oil, which is widely used in the paint industry. The oil produced from the plant source is also used for human consumption in many countries due to polyunsaturated linoleic acid, monounsaturated oleic acid, and steric acid that adds to the nutritional value [173,175,176,177]. The in-depth chemical analysis of safflower oil revealed its exponential nutraceutical benefits in the human diet due to its rich content of polyphenolic components. Studies reveal that dietary inclusion of the oil benefits the overall boosting of the immune system, improves and promotes skin and hair growth, curtails degenerative bone diseases, has cardioprotective and hepatoprotective effects, and has an anti-cancerous effect [176]. In addition, anti-elastase and anti-collagenase activity shown by the oil extracted from safflower reveals its use as a potent antiaging agent. 

Several reports have also revealed the use of safflower across the globe as constituents in several traditional medicinal applications, cosmetics, as a flavoring agent, as unguents, and as dyes. Safflower extracts have been used traditionally in various countries in treating various ailments such as arthritis and mastalgia, easing hypertension, treating colic, easing menstrual pain, and increasing fertility in both sexes [173,177]. The water extract of safflower has been reported to be used to reduce menstrual cramps, as laxatives, and as anti-inflammatory preparation in traditional concoctions [177,178]. It has antipyretic, analgesic, and purgative activity [173].

Safflower is also known as fake saffron or dyer’s saffron, and the flowerets are extensively used as a flavoring and coloring agent in Italian, British, and French cuisine. The carpeting industry across Eastern Europe, the Middle East, and the Indian subcontinent widely uses the flower as a dyeing agent due to its capacity to produce a wide range of intensity of colors from white to orange and reddish-orange pigmentation, determined by the presence of cardamine, a bioactive component reported for natural antioxidant capacity [173,177,179]. 

Brewed florets of safflower or safflower tea are a typical traditional drink in China and are widely used as a tonic tea for general health benefits and gastric and reproductive illness [173]. Tea made from the seed and flower extracts from the plant has also been shown to reduce metabolic dysfunction associated with alloxan-induced diabetes in rats. The hypolipidemic effect exhibited by the extract is due to its ability to promote antioxidant activity and reduce and maintain serum triglycerides and cholesterol levels, which in turn protects the beta cells and stimulates and enhances the production of insulin [179,180,181]. The rich phytochemical composition of safflower can be grouped into flavonoids (such as derivatives of quercetin, luteolin, acacetin, and apigenin), alkaloids, lignans, steroids, phenolics, quinochalocone C-glycosides, quinone containing chalcones, proteins, minerals, and polysaccharides [182]. Continued supplementation of safflower tea in the diet revealed its effectiveness in promoting bone formation and protection against osteoporosis in post-menopausal women via an increase in antioxidants such as α- tocopherol and ascorbic acid, a co-factor in serum that is necessary for bone remodeling [183,184].

Hydroxysafflor yellow A (HYSA), a primary water-soluble compound that is a natural pigment isolated from the dried florets of safflower, has shown a commendable neuroprotective effect against cerebral ischemia, dementia, Parkinson’s disease, and traumatic brain injury in various studies. Sun et al., (2018) reported the protective effect of HYSA by cerebrovascular dilation that helped in the partial restoration of cerebral flow in the ischemic region. The disrupted blood–brain barrier (BBB) permeability was also improved by HYSA [185]. It was also shown to alleviate platelet aggregation, blood viscosity, and thrombosis formation, which directly stops thrombin formation, which triggers post-ischemic cascade leading to neuro dysfunction, thus imparting neuroprotection. The reactive oxygen scavenging activity of HYSA provides direct protection against oxygen-glucose deprivation-induced injury in PC12 cells by suppressing neuronal oxidative stress and by the modulation of intrinsic apoptotic mitochondrial pathways [186,187]. Post cerebral ischemic conditions such as the excessive release of glutamate result in excitation injury in neurons due to overactivation of N-methyl D-aspartate receptors (NMDARs) and calcium overload resulting in ROS generation and mitochondrial membrane dysfunction are ameliorated by HYSA [187,188,189,190,191]. It was also found to provide a protective effect to dopaminergic neurons in PD mice models. Additionally, it was found to improve neuronal viability by reducing apoptosis and increase in antioxidant capacity [187]. 

HYSA also observed overall cardiovascular protection both in vitro and in vivo. It has been observed to provide an antihypertension effect obtained through the modulation of arterial pressure and heart rate [187]. In vitro studies revealed suppression of adventitial fibroblast proliferation and collagen synthesis stimulated by angiotensin II [187,192], promoting the inhibition of dedifferentiation and proliferation of vascular smooth muscle cells exhibited during hypertension [193]. Myocardial damage alleviation and promotion of angiogenesis were also observed during treatment with HYSA in rats [194]. Studies also reported a decrease in the levels and activity of malondialdehyde, reactive oxygen species, lactate dehydrogenase, creatine kinase-MB, and the protection of mitochondrial function and membrane potential [187,195]. 

Furthermore, the aqueous and methanolic extracts containing safflower yellow have also been reported to show potent anti-inflammatory activity via the suppression of NF-κB, which results in inhibition of the expression of Nrf-2, iNOS, and cox-2 gene regulation, leading to alleviation of pulmonary fibrosis. In contrast, HYSA has also been reported to induce anti-inflammatory effects by reducing pro-inflammatory cytokine and inflammatory cell infiltration during pulmonary edema and respiratory dysfunction. Other constituents that also show anti-inflammatory effects are quinochalcones such as ginkgolide B and saffloquinoside [196]. 

Recent studies have shed light on autophagy-mediated protective effects imparted by phytophenolic compounds from safflower extract, increasing its therapeutic value against cardiovascular and neurological impairment and metabolic dysfunction such as diabetes. As discussed in the previous sections, autophagy is a mechanism that allows the organism to maintain its normal homeostasis and is reported to play a crucial role in regulating the metabolism and modulation of the immune system [197]. Several essential pathways such as Wnt1-inducible signaling pathway protein 1 (WISP1), phosphoinositide 3-kinase (PI3K), protein kinase B (Akt), β-catenin, and mammalian target of rapamycin (mTOR) govern apoptotic and autophagic pathways during stress conditions related to disease or injury [198]. Jiang et al., (2017) reported the suppression of the PI3K/AKT/mTOR signaling pathway via the combinatorial effect of HSYA as a sonsosensitizer and sonodynamic therapy-induced autophagy through ROS production clearing atherosclerotic plaques and consequent protective effect [199]. It was also found that HSYA effectively reversed cellular and molecular changes and suppressed apoptosis induced by oxygen-glucose deprivation in brain microvascular cells (BMECs) by activating the PI3K/Akt/mTOR signaling pathway. The number of autophagosomes was also considerably reduced after exposure to the compound. The LC3 and Beclin-1 mRNA and proteins were also found to be downregulated. In a recent study by Li et al., (2019), HSYA in a dose-dependent manner inhibited the heat-stress induced mitochondrion-dependent apoptosis and autophagy in neural stem cells via the p38/MAPK/MK2/HSP27-78 signaling pathway [200]. 

### 2.3. Ginger Tea

*Zingiber officinale*, or ginger, belonging to the Zingiberaceae family, is a popular spice globally and commonly in Asian countries. Ginger has been used as a traditional household remedy to enhance digestion and alleviate gastrointestinal problems such as bloating, flatulence, loss of appetite, morning sickness, and heartburn, and has been a part of alternative medicinal fields such as Ayurveda and Chinese traditional medicine since time immemorial. It has been consciously used as a treatment against cold and flu due to its efficacy in boosting circulation and providing warmth to the body [201,202,203]. Consumption of anything more than necessary can accumulate in the body and lead to harmful effects. Therefore, knowing the phytocomponents of any herbal therapeutics is very important. The bioactive components of ginger are phenolics such as gingerols, shogaols, and paradols and terpenes such as β-bisabiolene, α-cucrcumene, zingiberne, α-farnesen and β-sequiphellandrene. Apart from these, many other phenolic compounds are present, such as quercetin, zingerone, gingerenone-A and 6-dehydrogingedione, as well as polysaccharides, lipids, organic acids, and fiber [203,204,205,206]. 

Various studies have evaluated ginger’s potential therapeutic and preventive benefits in treating a variety of pathological conditions. It has shown excellent pharmacological activities against oxidative stress, inflammation, cancer, gastrointestinal ailments, diabetes, nausea, cardiovascular disease, cholesterol, and neurological disorders. Studies have shown that oxidative stress disrupts antioxidant balance, leading to overaccumulation of reactive oxygen species (ROS), which results in macromolecular damage and cytotoxicity. Abolaji et al., (2017) showed alleviation of oxidative stress and inflammation induced by chlorpyrifos (CPF), an acetylcholine inhibitor that stimulates proinflammatory player’s ROS upon treatment with a 6-gingerol rich fraction [207]. The study also revealed the prevention of elevation of CPF induced nitic oxide and TNF-α in rats. Several other studies also reported its efficacy in stimulating the production of several antioxidants such as glutathione peroxide, glutathione reductase, and glutathione S-transferase and superoxide dismutase [208].

Several in vitro and in vivo studies have shown anti-tumorigenic and anticarcinogenic activities by activating the immune system and suppressing angiogenesis and metastasis, thus preventing cancers from being malignant [209,210,211]. Ginger is effective against various cancer types, including gastrointestinal cancer, breast cancer, lymphoma, colon cancer, skin cancer, hepatoma, and liver cancer. The anticancer activity of ginger can also be attributed to its antioxidant capacity [211,212]. Several studies demonstrated the cytotoxic effect of bioactive from ginger on various cancer types by promoting apoptosis, negatively regulating the expression of genes involved in PI_3_K/AKT and ERK pathway. A study by Tahir et al., (2015) revealed the promotion of apoptosis in colorectal cancer cells via down regulation of PI_3_K/AKT and ERK pathway followed by increased expression of caspase 9. In a pilot study, supplementation of the ginger extract showed a reduction in the expression of telomerase reverse transcriptase (hTERT) and MIB-1 and an increase in anti-apoptotic gene bcl-2 associated X (Bax) [213]. In another study, ginger supplementation indicated potential preventive action against colorectal cancer by inducing a reduction in the expression of cyclooxygenase-1 (COX-1), an important enzyme in the production of prostaglandin [203,214,215,216]. Studies reveal that the attenuation in the proliferation of prostate cancer cells in vitro and in vivo was mainly through the negative regulation of multidrug resistance-associated protein 1 (MRP1, a glutathione S transferase) by suppressing signal transducer and activator of transcription (STAT3) and NF-κB signaling. The expression of cyclin D1, survivin, c-Myc, and bcl-2 was found to be decreased during treatment with bioactive ginger extract [203,217,218]. Bioactive components from ginger, such as 6-shogaol, 6-gingerol, 10-gingerol, and 10-shogaol, were found to inhibit cell proliferation and induce apoptosis in breast, ovarian, liver, pancreatic, and cervical cancer cells by targeting cell cycle regulatory proteins cyclin A and D1, AMPK (5′adenosine monophosphate-activated kinase), mTOR signaling, and inducing caspases and p53 [203,219,220,221,222]. 

Several studies revealed ginger’s neuroprotective and anti-inflammatory effect and its bioactive components 6-shogaol, 10-gingerol, and 6-dehydrogingerdione. In vivo analysis showed its protective effects against Alzheimer’s disease and Parkinson’s disease by improving memory impairment induced by amyloid β1-42 plaque and alleviating cognitive dysfunction. The ROS scavenging capacity and inhibition of proinflammatory gene expression attribute to the anti-inflammatory effect shown by ginger [203,223,224]. 

Furthermore, the capacity to reduce the total cholesterol LDL, triglycerides, and very-low-density lipoproteins (VLDL) in high-fat-diet rats demonstrated a protective effect on cardiovascular function. Studies also demonstrated the promotion of vasodilation and prevention of platelet aggregation in hypertensive rats [203,225]. Other studies also reported the effectiveness of ginger against obesity and diabetes. 6-gingerol was found to enhance the tolerance of type 2 diabetic mice to glucose, increase the activity of glycogen synthase 1 and increase cell membrane presentation of glucose transporter type 4 that facilitated the storage of glycogen in skeletal muscles [203,226]. 

Studies have also revealed autophagy-mediated prevention and suppression of metabolic disorders with supplementation of bioactive components from the ginger extract. Several studies and reviews have extensively shown the importance and crosstalk between autophagy and apoptosis mechanisms vital for therapeutic approaches against cancer. Woźniak et al., (2020), reported apoptosis and autophagy during combinatorial application of 6-shogaol and other FDA chemotherapeutic agents, which resulted in enhanced efficacy in treatment against colorectal cancer [227]. Another study showed the induction of a protective effect by 6-gingerol via the activation of autophagy in HUVEC (human umbilical vein endothelial cells) against apoptosis through the suppression of the PI_3_K/AKT/mTOR signaling pathway along with an increase in Beclin1 and bcl-2, to induce autophagy and inhibit apoptosis, respectively [228]. A recent study demonstrated inhibition of lung cancer in vitro and in vivo by 6-gingerol via the autophagy-ferroptosis pathway [229]. SSi-6, a semi-synthetic molecule of 6-gingerol, reported cytotoxicity against triple-negative breast cancer via ROS-mediated caspase-independent manner, resulting in autophagy and apoptosis [230]. Ginger can be called a super food as it provides wholesome protection against most aliments from simple flu to severe metabolic disorders. The supplementation of ginger in our daily life as tea or food additives can enhance immunity and provide comprehensive health benefits.

### 2.4. Hibiscus Tea 

Among the many species of the hibiscus plant, the flower of Roselle (*Hibiscus sabdariffa* L.) is widely used to prepare hibiscus tea. Roselle is used in folk medicines. The potential properties such as anti-obesity [231,232,233], autophagy-inducing activity [234], antioxidant, anti-inflammatory, anti-hypertensive, anti-hyperglycemic properties indicate the health-promoting and protective health benefits of Roselle [234,235]. 

A randomized, double-blind trial using Roselle flower extract (dose: 2 capsules after meals, thrice a day for 12 weeks; 450 mg of Roselle flower extract and 50 g starch per capsule) in adults with BMI ≥ 27 kg/m^2^ (aged 18 to 65) showed a reduction in abdominal fat, serum-free fatty acids, obesity and exhibited improvement in the liver steatosis in obese adults [232]. Consumption of roselle tea twice a day for one month showed a positive effect on type II diabetic individuals’ blood pressure [236]. Meta-analysis studies by Serban et al., (2015) and Najafpour Boushehri et al., (2020) showed a significant association between roselle tea consumption and a reduction in both diastolic and systolic blood pressure, indicating the potential anti-hypertensive effect of the supplementation of roselle tea [237,238]. Aqueous extract of *H. sabdariffa* inhibited lipid accumulation and adipocyte differentiation in 3T3-L1 cells by inhibiting ROS generation and could manage metabolic disturbance and insulin resistance [231]. Roselle flower extract has been reported to suppress the expression of inflammatory cytokines, including IL-6 and TNF-α in dextran sodium sulfate-induced colitis in mice [239].

In the *C. elegans* model of Alzheimer’s disease, the flower extract has been shown to prolong the lifespan and protect against Aβ-induced paralysis by modulating DAF-16, SKN-1 mediated pathways [240]. The nutraceutical product *pres phytum* comprising *Olea europaea* and *H. sabdariffa* protected SH-SY-5Y cells from H_2_O_2_-induced oxidative damages by suppressing ROS generation and inhibiting apoptosis. Furthermore, the nutraceutical also exhibited brain penetration ability which could be corroborated by the neuroprotective property [241]. In ovariectomized Wistar rats, *H. sabdariffa* extract improved spatial memory and induced BDNF expression significantly compared to the control, indicating that the plant can act as a phytoestrogen and be used as alternative hormonal therapy [242]. Roselle extract showed a neuroprotective effect against streptozotocin-induced AD by inhibiting amyloidogenic pathway, suppressing the expression of pro-inflammatory cytokines, and downregulating p-p38 [243]. 

Roselle extract showed anti-apoptotic effects against monosodium glutamate-induced testicular damage by improving the antioxidant enzymes SOD, catalase and GSH in rats [244]. *H. sabdariffa* extracts protected human umbilical vein endothelial cells (HUVEC cells) against oxidized low-density lipoprotein (o-LDL) toxicity by downregulating the expression of caspases, Bax and inducing autophagy-mediated through the upregulation of class III phosphoinositide 3-kinase (PI3K)/Beclin-1 and Phosphatase and tensin homolog (PTEN)/class I PI_3_K/Akt cascade signaling proteins [245]. 

In human melanoma cells, *H. sabdariffa* extract exhibited cytotoxic effects mediated through autophagic machinery. The extract induced the expression of PI_3_K class III, ATG9, Beclin1, *Atg*-12, *Atg*-5, LC3 while downregulating the negative modulator of autophagy, p-AKT, and mTOR [246]. The extract was found to selectively induce apoptosis in both triple-negative and estrogen-receptor-positive MCF-7 and MDA-MB-231 cells in a concentration-dependent manner by inducing ROS level and loss of MMP [247]. A similar cytotoxic effect was also reported with delphinidin-rich extract of *H. sabdariffa*, in which the extract triggered an autophagic response by modulating the expression of LC3-II, AMPK phosphorylation [248].

### 2.5. Lemongrass Tea

*Cymbopogon citratus*, commonly referred to as lemongrass, is widely found in tropical regions and has been shown to have several medicinal properties, with citral as the active constituent. Lemongrass leaves contain phenolic compounds, and the leaves are used to prepare medicinal tea and essential oil. Supplementation of *C. citratus* for 30 days in human volunteers exerted an erythropoiesis boosting effect, attributed to the nutritional constituents [249]. 

In a lithium-pilocarpine epileptic rat model, citral administration increased epileptic latency, improved brain-derived neurotropic factor expression, and inhibited inflammatory cytokines including TNF-α, IL-6, NF-κB [250], and the activity was partly dependent on GABAergic neurotransmission [251]. Essential oil of lemon grass showed a neuroprotective effect against AlCl_3_ by improving motor functions, regulating antioxidant status, modulating acetylcholinesterase activity, and reducing Aβ levels [252,253]. The polysaccharide fraction, aqueous extract from *C. citratus* and citral inhibited LPS-induced neuroinflammatory response in RAW264.7 cells by downregulating the expression of IL-1β, IL6, NF-kB and TNFα against LPS-induced neuroinflammatory response [254,255,256]. The essential oil of lemongrass showed anti-anxiolytic activity by activating GABA_A_ receptor–benzodiazepine complex in mice [257]. Further aqueous extract of the extract exhibited anti-depressant-like activity mediated by the monoaminergic pathway, indicating the neuroprotective efficacy of the plant [258].

Apart from neuroprotection, *C. citratus* exhibits antiproliferative effects against various cancer cells. Ethanol extract of lemongrass showed a potent anti-cancer effect against colon and prostate cancer by inducing ROS level, causing depolarization of mitochondrial membranes, and reducing tumor burden and formation in mice [259,260,261]. The active constituent of lemongrass, citral, exhibited an anti-proliferative effect against prostate cancer cells by inhibiting lipogenesis. Citral treatment activated AMPK phosphorylation, thereby inhibiting the enzymes (SREBP1, HMGR) involved in fatty acid and cholesterol biosynthesis and leading to apoptosis by the upregulation of Bax [262]. Polysaccharide-rich fraction of the plant showed an anti-tumorigenic effect in mice by exhibiting immunomodulatory effect [263] and activated apoptotic factors in LNCap cells [264]. Essential oil from *C. citratus* also exhibited antitumor activity in 7,12-dimethylbenz-[α]-anthracene-induced breast cancer in rats, which could be attributed to the presence of isocitral, geraniol and myrcene [265]. In female Balb/c mice, essential oil administration showed a protective effect against N-methyl-N-nitrosurea-induced mammary carcinogenesis by inducing DNA damage in peripheral leucocytes [266]. In A549 lung cancer cells, essential oil of lemongrass induced cell cycle arrest and apoptosis by upregulating the expression of Bax [267]. 

In streptozotocin-induced diabetic rats, *C. citratus* mitigated endoplasmic reticulum stress by downregulating the expression of Grp78, PERK, ATF4, and CHOP, while activating the Nrf-2 pathway [268]. Furthermore, lemongrass extract also showed hepatoprotective effects against H_2_O_2_, carbon tetracholride-induced liver toxicity [269,270], and H_2_O_2_-induced reproductive system injury in rats by augmenting the antioxidant status [271]. Lemon grass polyphenols have been reported to protect HUVEC cells from high glucose, hydrogen peroxide, and oxidized low-density lipoprotein-induced oxidative damage by reducing ROS and NO levels [272]. Although there is no direct evidence suggesting that autophagy is mediated by the effects of lemongrass tea, the anti-cancer activity and activation of Nrf-2 suggest that they could mediate autophagy, as corroborated by the effects exhibited by green tea. However, more research is required to confirm this effect.

### 2.6. Rhinacanthus Tea

*Rhinacanthus nasutus* L., commonly known as snake jasmine, is used in traditional medicine to treat various disease conditions in Southeast Asia. The leaves of the plant are used to make herbal teas, commonly used to treat hepatitis, diabetes, and hypertension [273]. Leaf extracts were reported to have anti-fungal and anti-microbial activity against *S. mutans*, *S. epidermidis*, *P. acnes*, and *S. aureus* [274,275]. Overall, the plant and its active component, rhinacanthins, together can impart anti-cancer and neuroprotective effects along with antioxidant potential [273].

Rhinacanthins can induce antioxidant activity in hypoxia and β-amyloid-induced toxicity in neuronal cell lines [276,277] and attenuate neuroinflammation induced by LPS, Aβ, and interferon-γ in BV-2 and primary rat glial cells [278]. It could also benefit patients who have suffered from delayed ischemic neurologic deficit following an aneurysm by preventing neuroinflammation [279]. Rhinacanthins also displayed potent cytotoxic and antimetastatic activity against cholangiocarcinoma cells by inhibiting invasion, regulating focal adhesion kinase, and interfering with the MAPK pathway [280]. They were also cytotoxic to MCF7 breast cancer cells, the NCI-H187 cell line, and Vero cells [281]. Additionally, rhinacanthins could significantly trap methylglyoxal, thereby inhibiting the formation of advanced glycation end products showing anti-diabetic property [282].

Currently, there is no published direct evidence to suggest the autophagy properties of rhinacanthus tea. However, as we discussed in the previous sections, autophagy can play a significant role in anti-cancer, neuroprotection, and diabetes regulation. Currently, unpublished data from our laboratory has suggested that rhinacanthin-C can alter LC3-I/II expression in HT-22 cells. Furthermore, autophagic cell death has been hinted at in the anticancer effects of rhinacanthin-C [283].

### 2.7. Jasmine Tea

*Jasminum* sp., belonging to the Oleaceae family with fragrant flowers, is commonly grown in Asian and Mediterranean region. The essential oil of jasmine is used in traditional medicine for gastric spasms and cardiovascular diseases [284]. It is also used in aromatherapy as a rejuvenator and stress relaxant [285]. Jasmine green tea (JGT) is a type of green tea flavored with jasmine flowers. Few studies have been reported investigating the beneficial effect of JGT. Chan et al., (1999) investigated the hypolipidemic property of JGT water extract and epicatechins extracted from it using a high-fat diet (a diet containing 200 g lard per kg and 1 g cholesterol per kg) fed hamsters as an experimental model. Both extract (15 g per L) and epicatechins (5 g per L of distilled water) supplementation showed reduced serum levels of total cholesterol and triacylglycerols in high-fat-diet fed-hamsters [286]. These hamsters supplemented with different concentrations of epicatechins (1.1 to 5.7 g per kg of diet) showed a dose-dependent hypolipidemic effect. The time-course experiment in this combination showed a higher amount of total fatty acids and sterols (neutral and acidic) in their feces, indicating that the hypolipidemic effect of epicatechins is associated with the inhibition of absorption of dietary fat and cholesterol [286].

Ethanol extract of *Jasminium sambac* showed vasorelaxant effects in a concentration-dependent manner in rat aorta rings by inducing nitric oxide release, activating potassium channels, and inhibiting the influx of calcium ions and release of the same from the sarcoplasmic reticulum, which could be attributed to the high flavonoid glycoside and iridoid glycoside contents [287]. Even though there is no direct evidence suggesting the autophagy mediating effects of jasmine tea, the high flavonoid content and the hypolipidemic effect showed by epicatechins suggest that they could also mediate autophagy. However, more research is required to confirm this effect.

## 3. Negative Effects of Excessive Tea Consumption

Despite the positive effects of the many kinds of teas and their benefits in autophagy activation, several adverse effects can be seen, especially if tea is consumed in excess. 

### 3.1. Iron Deficiency Anemia

Iron deficiency anemia is a significant problem worldwide [288]. Excessive tea consumption interferes with iron absorption and can lead to iron deficiency anemia. One case study identified a 48-year-old type II diabetic male who suffered from iron deficiency. Upon cessation of green tea consumption, his iron deficiency condition improved. However, upon resumption of tea drinking, his hemoglobin levels began to decrease again [289]. The effects of teas on heme-iron absorption have been known for many years [290,291]. These iron-reducing effects have been attributed to the iron-binding effects of catechol-group-containing compounds found in teas [292]. However, unless consumed in excess, there is only a low risk of iron deficiency anemia unless the individual is already at risk of anemia [293].

### 3.2. Caffiene Consumption

The average cup of tea contains 30 to 60 mg of caffeine, which is less than the average cup of coffee, which contains around 100 mg of caffeine. However, tea drinks are often consumed more often than coffee, especially when tea is being used as a weight-loss supplement [294], leading to dangerous levels of caffeine consumption. Furthermore, caffeine consumption from tea in older women may predispose them to osteoporosis [295]. Although the risk of caffeine consumption is relatively low, primarily due to caffeine irritant effects on the gastrointestinal system leading to spontaneous emesis, deaths have been reported. Many deaths, however, are the result of the consumption of diet pills or sports drinks rather than tea [296].

### 3.3. Tea and Cancer

Teas are often thought of as having anti-cancer effects; however, several studies indicate the opposite. For example, the consumption of hot beverages, including hot tea, has been associated with an increased risk of esophageal cancer [297], particularly when combined with tobacco smoking and alcohol [298]. However, some studies showed that the consumption of green tea was preventative against esophageal cancer [299,300].

Some studies have identified black tea drinking as a potential risk factor for stomach cancer, although these studies have some confounding factors such as inappropriate controls, and when controlled for age and socioeconomic status, the link is no longer significant [301,302].

## 4. Conclusions

It is now well recognized that aberrant regulation of autophagy is among the critical hallmarks of several pathological conditions, including common worldwide health problems such as cancer, diabetes, obesity, and Alzheimer’s disease. Regardless of its paradoxical role, autophagy represents an attractive therapeutic target in the fight against those diseases. In this review, six of nine different popular herbal teas, including green tea, oolong tea, black tea, safflower tea, ginger tea, and hibiscus tea, as well as their polyphenolic components (e.g., EGCG, HSYA, 6-gingerol, and delphinidin) have shown their capacities in the modulation of autophagic pathways, mainly through the activation of the PI3K/AKT/mTOR signaling pathway, which could correspond to their reported health-promoting and therapeutic properties. Although there is currently no published direct evidence for the autophagy-mediating effects of the other three types of herbal tea (lemongrass tea, rhinacanthus tea, and jasmine tea), the previous reports of their health benefits hint an involvement of autophagy in the underlying mechanisms which should be researched further. As the popularity of herbal tea consumption has markedly increased worldwide and almost every individual takes tea as a part of their diet, it is essential to understand more clearly about their possible effects on our health, particularly on maintaining a balance in the level of autophagy, where both too much and too little can be harmful.

Furthermore, it is essential to remember that any tea infusion (hot water extract) may not necessarily extract all the bioactive compounds found and extracted with other polar or non-polar solvents from the leaves of these plants. Moreover, these compounds may be relatively dilute, meaning that a high level of tea consumption might be needed to attain the necessary bioactive compounds to impart health benefits. However, this itself comes with safety and efficacy issues, which have to be closely monitored. Therefore, further studies on clinical trials should be done to identify the proper balance needed to provide the maximum effect.

## Data Availability

Not applicable.

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
