# Peer review of "Role of Herbal Teas in Regulating Cellular Homeostasis and Autophagy and Their Implications in Regulating Overall Health"

_nutrients, 2021, doi:10.3390/nu13072162_

Round 1

Reviewer 1 Report

­­The manuscript presents information about bioactive compounds of herbal teas and their influence on human health. The paper is well structured and written. The authors collect a lot of references to support the topic. The literature cited is closely connected and appropriate to the topic. But in my opinion title is not clearly reflected in body of manuscript. Many presented roles of herbal teas bioactive ingredients are not directly connected with autophagy, but rather with human health in general. Maybe the authors will consider changing of title of manuscript. Another issue is that the authors sometimes discuss compounds and properties of plants extracts. These compounds can be extracted by methanol or ethanol and maybe not be present in teas, which are water extract. Also tea infusion is very diluted, so content of bioactive compounds could be very low compare to extract.

Line 760 – 761

Do you have any data/references to support this statement?

Author Response

Comment: The manuscript presents information about bioactive compounds of herbal teas and their influence on human health. The paper is well structured and written. The authors collect a lot of references to support the topic. The literature cited is closely connected and appropriate to the topic. But in my opinion title is not clearly reflected in body of manuscript. Many presented roles of herbal teas bioactive ingredients are not directly connected with autophagy, but rather with human health in general. Maybe the authors will consider changing of title of manuscript.

Response: Thank you for your appreciation and suggestions. We have modified the title to be “Role of Herbal Teas in regulating cellular homeostasis and autophagy and their implications in regulating overall health” to better reflect the content of manuscript which describes the roles in both autophagy and general health aspects.

Comment: Another issue is that the authors sometimes discuss compounds and properties of plants extracts. These compounds can be extracted by methanol or ethanol and maybe not be present in teas, which are water extract. Also tea infusion is very diluted, so content of bioactive compounds could be very low compare to extract.

Response: We sincerely thank the reviewer for this valuable comments. This point has now been covered and discussed in the conclusion on page 17, line 853-862.

Comment: Line 760 – 761 Do you have any data/references to support this statement?

Response: We have added the reference to support the sentence “Many deaths, however, are the result of the consumption of diet pills or sports drinks rather than tea.” on page 16, line 822-823.

Reviewer 2 Report

It's a good comprehensive review. Please include a small section describing importance of herbal teas in the resolution of inflammatory stress and autophagy. 

Author Response

Comment: It's a good comprehensive review. Please include a small section describing importance of herbal teas in the resolution of inflammatory stress and autophagy.

Response: Thank you for your appreciation and suggestions. According to your suggestion, we have added the section of 1.3.4 Inflammatory stress in this revised manuscript (page 2, line 70-73 and page 4, line 169-192). Also, this revised manuscript has been spell checked.